# Temporal Pattern Attention for Multivariate Time Series of Tennis Strokes Classification

**DOI:** 10.3390/s23052422

**Published:** 2023-02-22

**Authors:** Maria Skublewska-Paszkowska, Pawel Powroznik

**Affiliations:** Department of Computer Science, Lublin University of Technology, 20-618 Lublin, Poland

**Keywords:** sport, tennis strokes, human action recognition, A3T-GCN, motion capture

## Abstract

Human Action Recognition is a challenging task used in many applications. It interacts with many aspects of Computer Vision, Machine Learning, Deep Learning and Image Processing in order to understand human behaviours as well as identify them. It makes a significant contribution to sport analysis, by indicating players’ performance level and training evaluation. The main purpose of this study is to investigate how the content of three-dimensional data influences on classification accuracy of four basic tennis strokes: forehand, backhand, volley forehand, and volley backhand. An entire player’s silhouette and its combination with a tennis racket were taken into consideration as input to the classifier. Three-dimensional data were recorded using the motion capture system (Vicon Oxford, UK). The Plug-in Gait model consisting of 39 retro-reflective markers was used for the player’s body acquisition. A seven-marker model was created for tennis racket capturing. The racket is represented in the form of a rigid body; therefore, all points associated with it changed their coordinates simultaneously. The Attention Temporal Graph Convolutional Network was applied for these sophisticated data. The highest accuracy, up to 93%, was achieved for the data of the whole player’s silhouette together with a tennis racket. The obtained results indicated that for dynamic movements, such as tennis strokes, it is necessary to analyze the position of the whole body of the player as well as the racket position.

## 1. Introduction

Computer Vision is an interdisciplinary field of study that aims to derive meaningful information from various types of data. Applying artificial intelligence for digital images, skeleton, depth, videos, point cloud, audio, acceleration, signals or motion capture data allows one to perform actions or make decisions as well as further recommendations. The purpose of Human Action Recognition (HAR) is to understand human behaviours and identify them [1,2]. It specifies a set of person’s moves performed in time in order to complete a task. Occasionally, additional objects, such as a tennis racket or a golf club, are involved to do the actions. Depending on the complexity of the movements and their duration, different length sequences are taken into consideration, from a single frame to a whole video streaming. HAR is a challenging task used in numerous applications. It interacts with many aspects of Computer Vision, Machine Learning, Deep Learning and Image Processing [3]. It utilizes detection of a person or objects in the image, video as well as sensor data, the location of the action in time and space, and the recognition of the action. This attitude usually involves feature detection, such as extracted from 3D silhouettes, skeletal joint and body part location, local spatio-temporal, local occupancy patterns and finally 3D scene flow [2]. That is why it makes a significant contribution to sport analysis. Detection of athletes and recognition of their actions or teams’ activities plays a pivotal role in indicating the players’ performance level and training evaluation or analyzing sport statistics [3].

Classification is a challenging task; however, its use can be found in many studies considering various sport disciplines, for both individual and team ones. Twelve basketball activities, corresponding to the most of the complex actions, were recognised in [4]. For this purpose the LSTM-DGCN method was proposed. It consisted of two parts: Deep Graph Convolutional Network (DGCN) and Long Short-Term Memory (LSTM). Basketball players measures such as distances between joints and angles were input parameters. Moreover, selection coordinates and depth maps together with RGB frame sequences were used for this purpose as well.

Both action recognition and analysis of the karate athletes were presented in [5]. The Attention-enhanced Graph Convolutional LSTM Networks (AGC-LSTM) was applied for recognition of the actions gathered from five athletes. The analyzed movements such as a punch (middle, upper, forward upper and back upper), back pounce, kick (side rising, back, inside crescent, side) as well as several technical moves were taken into consideration. A framework dedicated to sports video recording considering attentive movement characterization was presented in [6]. It involved hierarchical recurrent neural networks. The extraction of human pose, use of trajectory clustering made it possible to describe a dynamic movement of players or the whole teams as well as various interactions in sport games, such as volleyball. The classification of water sports using the Convolutional Neural Network (CNN) with discriminative filter banks was presented in [7]. Water skiing and surfing were taken into consideration. Both 2D and 3D data were analysed. Three branches were applied for the classifier consisting of: the average pooling classification head, a set of convolutions, spatial upsampling and max-pooling layers.

Recently, a novel approach to analyzing the movements of athletes has been proposed. It took into consideration the natural connections between the joints of the human silhouette in order to apply the graphs. The use of Graph Convolutional Neural Networks (GNN) made it possible to capture existing patterns and dependencies embedded in a spatial configuration of joints, as well as their temporal dynamics and thus it has become a very popular method in the field of HAR [8]. In that study, the authors introduced the Spatial Temporal Graph Convolutional Networks (ST-GCN) for everyday activities, sports areas and actions recognition. The same network was applied for the recognition of the selected movements in competitive sports [9]. Activities such as balance beam, diving, athletics, boxing, keeping fit, and badminton were taken into consideration. The data were recorded with the use of the markerless motion capture system. In [10], it was stated that a model based on temporal convolutional networks was more appropriate for HAR than a model considering the recurrent neural ones. These two approaches were compared using the NTU RGB+D dataset containing various types of actions. In [11], a new Two-stream Adaptive Graph Convolutional Network (2s-AGCN) for skeleton-based action recognition was proposed, which was verified by two datasets NTU-RGBD and KineticsSkeleton. This approach included additional information of skeleton data, such as the bone location. This attitude enhanced the performance of the classifier. The ST-GCN for skeleton action recognition was also applied for a similar approach in [12]. The Actional-Structural Graph Convolution Network (AS-GCN) was proposed. Its structure was characterised by basic building blocks for indicating spatial and temporal features. This new classifier was verified with NTU-RGBD+D and Kinetics datasets. Various types of sports, such as: golf, kicking, lifting, diving, running, horse riding, skateboarding, swing-bench, and walking, were recognized using the Part-Attention Spatio-temporal Graph Convolutional Network (PSGCN) [13]. It exploited the dynamic information from a sports video.

Classification of the tennis movement may be found in many scientific papers. The studies involved only skeleton-based action recognition, only tennis racket position as well as the whole player silhouette together with a racket. The research concerning SensorTile attached to the tennis racket was presented in [14,15]. For the purpose of swing classification Deep Neural Network [14] was applied, while for topspins (forehand and backhand), subpar forehand, subpar backhand, and slices (forehand and backhand) recognition the following methods were used: Support Vector Machine (SVM), Neural Networks (NN), Decision Tree (DT), Random Forest (RF) and k-Nearest Neighbor (kNN) [15]. The Pan Tompkins algorithm for the classification of shots using time warping was presented in [16]. Many studies focused on tennis stroke recognition based on video data. This attitude involved extracting features from videos and applying a classifier to the whole set [17]. The THETIS is a very well-known dataset consisting of twelve tennis moves captured by Microsoft Kinect in a form of video and ONI files [18]. The video-based action recognition of backhand (two-handed, one-handed, slice, and volley), forehand (flat, open stance, slice, and volley), serve (flat, kick, and slice) as well as smash was performed using the 3-layered LSTM network in [19,20]. In [17], these twelve moves were classified using SVM and linear-chain Conditional Random Fields (CRF). The five-layer deep historical LSTM network described in [21] was applied for similar moves using the following datasets: THETIS and HMDB51. Six tennis strokes from the THETIS datasets were recognised by the LSTM network in [22]. Serve, hit as well as non-hit were recognized by the Kernelised Linear Discriminant Analysis (KLDA) in [23]. Transductive transfer learning for an annotation of video sequences was applied. The changes in the tennis ball were also taken into consideration. The basic tennis strokes, forehand and backhand, from a video were analyzed in [24,25,26] using the SVM classifier. In [27], tennis serves, forehand and backhand were recognised using two classifiers: SVM with the radial basis function kernel and K-Nearest Neighbour classifiers (KNNs). A wireless inertial measurement unit sensor together with a system consisting of eight video cameras was used for capturing the data.

Studies concerning HAR were also performed using motion capture data recorded via optical systems. Forehand and backhand strokes with and without ball contact as well as no-shots were recognized by ST-GCN based on images generated from three-dimensional motion data in [28]. Graph Convolutional Networks (GCNs) were an obvious choice due to the fact that the parts of the image correlated with the human topology. In this study, the influence of input fuzzification on the obtained accuracy was examined. The results showed that this approach increased recognition ability. An extension of the above research was the recognition of individual tennis stroke phases, i.e., forehand preparation, forehand shot with racket swinging, backhand preparation, backhand shot with racket swinging and no-shots which were presented in [29]. Three classifiers with and without fuzzification were taken into account: SVM, MLP, and ST-GCN. In addition, the influence of the extensions and generalizations of the Choquet integral on the aggregation of results obtained by individual classifiers was verified. The results indicated that this method increased the efficiency of recognizing tennis moves. Another approach to tennis movements recognition including its phases was presented in [30]. For the purpose of the classification, the Attention Temporal Graph Convolutional Network (A3T-GCN) was applied both with and without input fuzzification. The conducted results showed that this classifier might be considered as one of the most appropriate methods for tennis classification.

The state-of-the-art study presented in this paper is to apply the A3T-GCN classifier for tennis stroke recognition based on three-dimensional coordinates data obtained from the optical motion capture system. Forehand, backhand and volley strokes were taken into consideration. The main purpose of this study is to look into how the content of three-dimensional data influences classification accuracy, precision, recall, and F1 score. Both the coordinates associated with the player’s silhouette and the position of the racket were analyzed, which to the authors’ knowledge is the novelty approach. The A3T-GCN was chosen due to the attention model, which both stores information about the player’s model, but also determines the predicted player position.

The rest of this paper is organized as follows. Section 2 explores the material and methods as well as introduces the Attention Temporal Graph Convolutional Network. Section 3 presents results of the state-of-the-art action recognition methods with the proposed classifier and 3D motion capture data. Section 4 discusses the proposed method, and finally Section 5 concludes the study.

## 2. Materials and Methods

### 2.1. Participants

In this study, seven male and three female tennis players took part (age 23.7±4.58, height 1.77±0.13 m, weight 71.65±10.68 kg). Only one of them was left-handed, while the others were right-handed. They all signed the consent for the study.

### 2.2. Data Acquisition

Each participant was prepared for the experiment. First, they have a 15-min warm-up. Second, thirty-nine retroreflective markers, specified in the Plug-in Gait model, were attached to their body. Finally, all the required measurements were gathered for the purpose of creating a new model as well as preparing its calibration in the motion capture system. Furthermore, seven markers were also attached to the tennis racket, according to the following scheme: one to the top of the racket head, two on both sides of the racket, one to the bottom of the racket head and one to the bottom of the racket handle. Such an arrangement reflects the racket shape and capture its movements.

For the purpose of acquisition, eight-camera optical Vicon motion capture system, installed in the indoor room, was used with the Nexus software. The cameras are mounted two on each wall on the same level. The whole schema of the cameras arrangement is presented in Figure 1. Before movement acquisition the calibration of the system was performed. The maximal calibration error did not exceed 0.045 pixels. The frequency of capturing was set to 100 Hz.

Each participant performed forehand, two-handed backhand and volley strokes. Forehand and backhand ones were performed while running and avoiding a bollard placed on the floor. Due to this, the strokes were more natural than hitting the ball from a standing position. At first, ten forehand strokes without a ball were performed, followed by ten backhand strokes without a ball. Next, these exercises were repeated with a ball. Finally, the participant performed ten volley forehand and ten volley backhand in front of the tennis net. Tennis balls were thrown from the right and the left side of the net, while standing parallel to the net, the player made a short movement with the racket in front of him/her, causing the ball to bounce and fall. The participant hit a ball which was caught by a special net. The forehand tennis stroke is made with the dominant hand. The racket was placed on the dominant side; then, it was directed towards the ball. After the racket made contact with the ball, the racket was directed to the opposite arm of the player in a way of swinging. While performing a two-handed backhand stroke, the racket was held with a continental grip. It was placed on the opposite side to the dominant one. After the racket made contact with the ball, it was directed to the dominant side. In the case of a one-handed backhand, the racket was held with a dominant hand. These two types of strokes are presented in Figure 2. It is worth indicating that forehand and forehand volley are very similar moves in a certain part of the movement. The same goes for backhand strokes.

Each performed stroke has been verified by a specialist. All failed strokes were rejected. Due to the fact that professional tennis players participated in the study, the well-performed strokes were repetitive.

### 2.3. Data Post-Processing

The Vicon Nexus software was used for post-processig of all obtained recordings. This tasks involved the following steps: marker labelling, gap filling using interpolation methods implemented in Vicon Nexus software (Pattern Fill and Rigid Body Fill), data cleaning, and applying the Plug-in-Gait model. The last one was only for the model representing human body. Additionally, a new model, consisting of all markers attached to the racket, was generated. The data prepared in this way was saved to c3d file.

The whole gathered recordings was verified by a professional tennis coach. As a result, the following number of tennis moves was obtained: backhand—212, forehand—197, forehand volley—180, backhand volley—180.

### 2.4. Attention Temporal Graph Convolutional Network

The idea of the A3T-GCN was taken from the work [31], where a similar structure was used to predict traffic volume in selected cities. The basic modification of this network consists of transforming the element responsible for the prediction into a classifier. Additionally, the elements responsible for the separation of spatial and temporal features have also been adapted. In the original approach, the Gated Recurrent Unit (GRU) network was applied. Due to extensive structure of the GRU network, inadequate to the problem, we are analyzing in our work RNN network, often also called BiRNN or Bidirectional RNN. It is schematically shown in Figure 3. Moreover, the original prediction was based on a Context Vector, while in case of this study additional Multilayer Perceptron was added on the output of the classifier. The whole network structure used in this study is presented in Figure 4.

The input data is arranged in a way of a graph G=(V,E) consisting of *M* nodes. They denotes *M* joints and their position changes in time. The *M* value was equal to 39 for the study without a tennis racket and 46 in case of experiments with it. Each node is described as a set of three-dimensional values $V=\{v_{ti}|t=1,\ldots ,T,i=1,\ldots ,M\}$.

#### 2.4.1. Spatial Features

Usually, skeleton data studies are based on images or video as input, so the data are processed by typical Convolutional Neural Networks (CNN). In case of this study, as input data points in three-dimensional space were used, the proposed classifier was based on Graph Convolutional Networks (GCNs). The connections between the nodes of the *G* graph were presented in the form of the adjacency matrix *A*. The entire feature matrix has been marked with the *X* variable. To process graph nodes, the GCN network, uses a Fourier filter to determine the spatial relation between features. This relationship was characterized by Equation (Equation 1), which actually defines a multilayer GCN model.
(1)$$F^{\left(n+1\right)}=\sigma \left({\tilde{T}}^{-\frac{1}{2}}\tilde{O}{\tilde{T}}^{-\frac{1}{2}}F^{\left(n\right)}\Theta^{\left(n\right)}\right)$$
where *n* represents the number of hidden layers, $\tilde{O}=O+I_{N}$ is the adjacent matrix (*O*) with added self-connections, $I_{N}$ describes the identity matrix, $\tilde{T}=\sum_{j}{\tilde{O}}_{ij}$, $F^{\left(n\right)}$ defines the output of *n* layer, $\Theta^{\left(n\right)}$ is a matrix which contains all parameters of specified $n^{th}$ layer and σ(·) represents the sigmoidal function for a nonlinear model [32].

In this study, the GCN network consists of three layers. This structure can be described by Equation (Equation 2).
(2)$$f\left(I,O\right)=\sigma \left(\hat{O}ReLU\left(\left(\hat{O}I\Psi_{0}\right)\hat{O}I\Psi_{1}\right)\Psi_{2}\right)$$
where $\hat{O}={\tilde{T}}^{-\frac{1}{2}}\tilde{O}{\tilde{T}}^{-\frac{1}{2}}$ indicates the preliminary step, $\Psi_{0}\in R^{PxF}$ denotes the weight matrix between input and hidden layer, *P* defines the size of the feature matrix, while *F* is a value related to the number of the hidden unit, $\Psi_{1},\Psi_{2}\in R^{FxZ}$ define the weight matrices from hidden to output layer, $f\left(I,O\right)\in R^{NxZ}$, denotes the output length *Z* and ReLU(), is the Rectified Linear Unit, commonly used as neurons activation function [32].

#### 2.4.2. Temporal Features

To indicate temporal features, which are the key elements in recognizing the analyzed types of tennis strokes, a BiDirectional Recurrent Neural Network was used. BiRNNs were applied to obtain the information about the player at time *t*. To gather this kind of data, the information about previous (in time n−1, n−2,…$n-n_{f}$, where $n_{f}$ denotes the maximum number of frames in all c3d file) features were taken into consideration. If analyzed file had fewer frames the missing values were set to 0. The structure of whole temporal features elements can be expressed by Equations (Equation 3)–(Equation 6) [31]:(3)$$ugc_{t}=\sigma \left(W_{u}*\left[X_{t},h_{t-1}\right]\right)$$
(4)$$rgc_{t}=\sigma \left(W_{r}*\left[X_{t},h_{t-1}\right]\right)$$
(5)$$mc_{t}=\tanh\left(W_{c}\left[X_{t},\left(rgc_{t}*h_{t-1}\right)\right]\right)$$
(6)$$h_{t}=ugc_{t}*h_{t-1}+\left(1-ugc_{t}\right)*mc_{t}$$
where $ugc_{t}$ denotes the update gate, which role is connected with controlling the information quantity at the previous moment, $rgc_{t}$ indicates the reset gate, which is responsible for neglecting the state information at the previous moment, $mc_{t}$ describes stored memory content at the current moment and $h_{t}$ defines the output value at the current moment. $W_{u}$, $W_{r}$ and $W_{c}$ represent the weights in the training process for the updated gate layer, reset gate layer and output layer, respectively.

#### 2.4.3. Attention Model

Commonly attention model is defined as an encoder–decoder. It is widely used in such applications as: traffic forecasting [31], image labeling [33], recommendation systems [34] or document classification [35]. Based on [36], it can be stated that the most general division of that kind of model includes hard and soft attention. In this study, the soft one was applied. The attention model’s first application is to store information about the player’s model. The second is to indicate the context vector, which is responsible for determining the predicted tennis player positions. The applied attention model consists of the following steps.

(1)First, determine, using the BiRNN network, the successive hidden states $u_{k}\left(k=1,\ldots ,m\right)$ of the time series $I_{k}\left(k=1,2,\ldots ,m\right)$, where *m* is the number of frames in series. As a result, the set of $u_{k}$ states is defined.(2)Second, a context vector ($C_{v}$) is determined. In particular, the value of position change was determined on a basis of two hidden layers. Their features are indicated applying Equation (Equation 7):
(7)$$e_{i}=\psi_{\left(2\right)}\left(\psi_{\left(1\right)}H+b_{\left(1\right)}\right)+b_{\left(2\right)}$$
where $\psi_{\left(1\right)}$ and $b_{\left(1\right)}$ denote the weight and bias of the first layer and $\psi_{\left(2\right)}$ and $b_{\left(2\right)}$ are similarly features of the second layer. *H* is a matrix with hidden layer values. To determine the values of $\psi_{\left(1\right)},\psi_{\left(2\right)}$ the Softmax function (Equation 8) is used.
(8)$$Softmax=\frac{exp(e_{i})}{\sum_{k=1}^{n}exp\left(e_{k}\right)}$$Final, the $C_{v}$ is defined as follows:
(9)$$C_{v}=\sum_{i=1}^{n}Softmax*h_{i}$$

The final classification is performed by two-layer perceptron. This neural structure consists of one element in the first layer and four (related to four recognised strokes) in the second one. The Softmax function is used to activate the neurons in the first layer, while the second layer is activated by a linear one.

### 2.5. Experiment

In this study, the forehand, backhand, volley forehand and volley backhand strokes were recognized. The whole tennis movements dataset consisted of backhand—212, forehand—197, volley forehand—180, and volley backhand—180. It represented the player’s silhouette together with a tennis racket. Two types of experiments were performed. The first one concerned the whole set of data while the other only the player’s silhouette by removing the coordinations of the tennis racket subject.

A series of experiments were carried out, taking into account the random division of data into the training, validation and test sets: 60%, 20% and 20%, respectively. The data was chosen from every type of stroke in the above-mentioned proportions. For each division, 20 tests were carried out, independently.

## 3. Results

Grouped results were presented in Table 1, Table 2, Table 3, Table 4 and Table 5. Selected parameters of the learning process showing the correctness of the model were shown in Figure 5. The loss value $L_{CE}$ was calculated on a basis of the Sparse Categorical Cross-Entropy defined by Equation (Equation 10).
(10)$$L_{CE}=-\sum_{i=1}^{n}T_{i}log\left(p_{i}\right),$$
for *n* classes, where $T_{i}$ is a ground truth, $p_{i}$ is the Softmax probability for the *i*th class.

The assessment of the classifier quality was based on several standard measures, such as: Accuracy (Equation 11), Precision (Equation 12), Recall (Equation 13) and F1 score (Equation 14).
(11)$$Accuracy=\frac{Number\;of\;correct\;classifications}{Total\;number\;of\;classifications}$$
(12)$$Precision=\frac{TP}{TP+FP}$$
(13)$$Recall=\frac{TP}{TP+FN}$$
(14)$$F1=\frac{2*Precision*Recall}{Precision+Recall}$$
where TP denotes the true positive fraction, FP—the false positive fraction, and FN—the false negative fraction.

Although Precision, Recall, and F1 are usually presented for binary classification, there is a simple way for extending their definition to multiple classes. In this case, Precision, e.g., for backhand, will be defined as correctly classified backhand strokes out of all classified backhand strokes. The Recall for backhand is the number of correctly predicted backhand strokes out of all input backhand strokes.

For the obtained accuracy results for two types of moves: strokes without and with a tennis racket (Table 1), the *T*-Test was calculated, for which t=−8.2753 was obtained, for α=0.05. The obtained result allowed us to state that it cannot be concluded that there is a difference between the means.

In order to check the correctness of the developed model, Leave-One-Out Cross-Validation (LOOCV) was performed (Table 6). This is a computationally expensive procedure; however, it allows us to obtain clear and unbiased information about the model. Using LOOCV, the root mean squared error (RMSE) for *n* tests was determined:(15)$$MSE=\frac{1}{n}\sum_{i=1}^{n}{\left(y_{i}-\hat{y_{i}}\right)}^{2}$$
where *n*—denotes number of test, $y_{i}$—true value, $\hat{y_{i}}$—predicted value.
(16)$$RMSE=\sqrt{MSE}$$

## 4. Discussion

In this study, the recognition of tennis forehand, backhand, volley forehand and volley backhand was performed based on data gathered in c3d files in a form of three-dimensional coordinates. Both the player’s silhouette indicated by 39 markers and the silhouette together with a tennis racket represented by 7 additional markers were analyzed.

As it can be observed in Table 1 and Table 2 and Figure 6 the mean accuracy depends on the captured data. It is higher for the experiments with the whole tennis player’s silhouette together with a tennis racket than the experiment involving only a single body. It can be concluded that the arrangement and trajectory of a tennis racket plays an extremely important role in the correct classification.

Furthermore, in the case of Precision (see Table 3), the obtained mean results are higher for the combination of the tennis player’s body model with a tennis racket for all analyzed strokes. The same dependence applies to Recall results (Table 4) and the F1 score (Table 5).

It should be noted that the standard deviation, amounting to a few percent, is low for all the obtained results, which shows the stability of the applied classifier.

In Table 7, the state-of-the-art studies related to the tennis strokes recognition are presented. They were performed using various types of data obtained from different sources, such as sensors, video or motion capture systems. The most research in this field were carried out on the well-known THETIS database. Both the data in the form of video, as well as the images obtained from the Kinect motion capture system, were the source for recognizing tennis movements. Broadcast video involving real matches or tournaments with top tennis players was also often taken into consideration. Various types of neural network approaches were used for these purposes. It is worth stressing that graph neural networks (ST-GCN and A3T-GCN) were used to recognize basic tennis strokes based on data obtained from an optical motion capture system. This classifier was chosen due to the characteristics of the recorded data. The applied human model, represented by 39 markers attached to the body in fixed locations, is transformed into a graph, which reflects the topology of the human silhouette. This approach allowed us to obtain a high accuracy. Analyzing the results presented in the study [28], it can be seen that used the A3T-GCN classifier allows for better recognition of tennis strokes than the ST-GCN one, despite the different types of input data. In the previous study, described in [28], the accuracy was obtained at the level of 68.9% for the 60% of data belonging to the training set, which corresponds to the settings in this study. The forehand and backhand classification was performed using images containing the subjects of the tennis player together with the racket. They were generated based on three-dimensional data by Vicon Nexus software. Based on this kind of input data the simplified model, both for tennis player and a racket, was created. The tennis racket was represented only by two points referring to its head and handle. The mean accuracy obtained in this study is higher for both the analyzed silhouette and its combination with a tennis racket compared to the results obtained in the work [28]. In [30], the study concerned the images obtained from three-dimensional data were analysed. The classification of forehand, divided into preparation and the hit phases, and backhand, also divided into preparation and the hit phases, as well as no-hit was performed using the Attention Temporal Graph Convolutional Network. The achieved accuracy results in a form of mean of two phases for forehand stroke did not exceed 80% while for backhand—77%. The obtained mean accuracy results in this paper are higher for the analyzed strokes with and without a tennis racket. The studies presented in this paper concerning tennis movements recognition was performed based on three-dimensional data in the form of coordinates of markers placed on the player’s body and a tennis racket was used. As it can be seen in Table 7 this approach is unique.

The results obtained in this paper suggest that the type of input data affects the accuracy of tennis stroke classification. Whole body data stored in the form of three-dimensional coordinates allows to achieve better results than in the case of images obtained from three-dimensional data. In addition, the inclusion of a tennis racket in the input data improves the classification quality of these strokes.

## 5. Conclusions

The state-of-the-art of this study was to verify the impact of adding a tennis racket to the input data of the whole player’s silhouette on the final classification of four main tennis strokes. For the purpose of this study the A3T-GCN classifier was applied. The tennis moves were represented in a form of three-dimensional motion data. The described approach gave satisfactory results. Forehand, backhand, volley forehand and volley backhand were taken into consideration. According to the previous authors’ study considering the ST-GCN network as a classifier [28], the obtained accuracy for two basic tennis movements (forehand and backhand) recognition in this research has been improved. As the results showed, adding a tennis racket to the data presenting the whole body silhouette significantly improved the classification quality. While comparing two graph networks, i.e., A3T-GCN and ST-GCN, the obtained results in this study clearly indicated that the A3T-GCN structure might be considered as the most suitable for motion data expressed in a form of three-dimensional coordinations. Further work will focus on the possibilities of using other methods of computational intelligence, in particular deep learning methods, and investigating their impact on the efficiency of movement classification in sport. Moreover, research on the possibility of using aggregation of classification methods, may be of interest in further studies in this field.

## Figures and Tables

**Figure 1 sensors-23-02422-f001:**
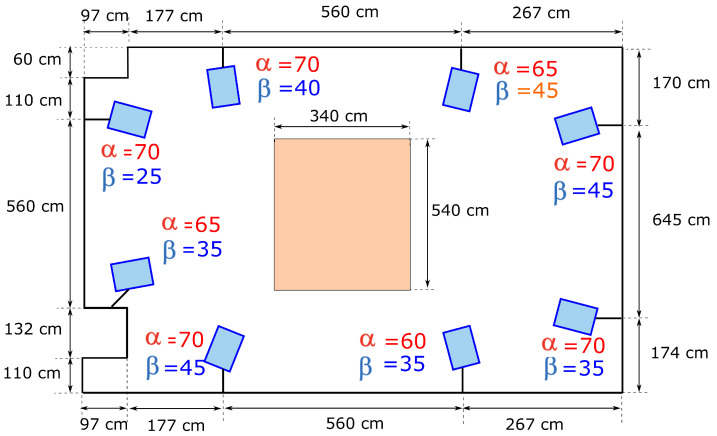
Motion capture cameras arrangement, where α is the angle in the OX plane between the floor and the camera axis, β is the angle in the OY plane between the camera axis perpendicular to the floor and the camera.

**Figure 2 sensors-23-02422-f002:**
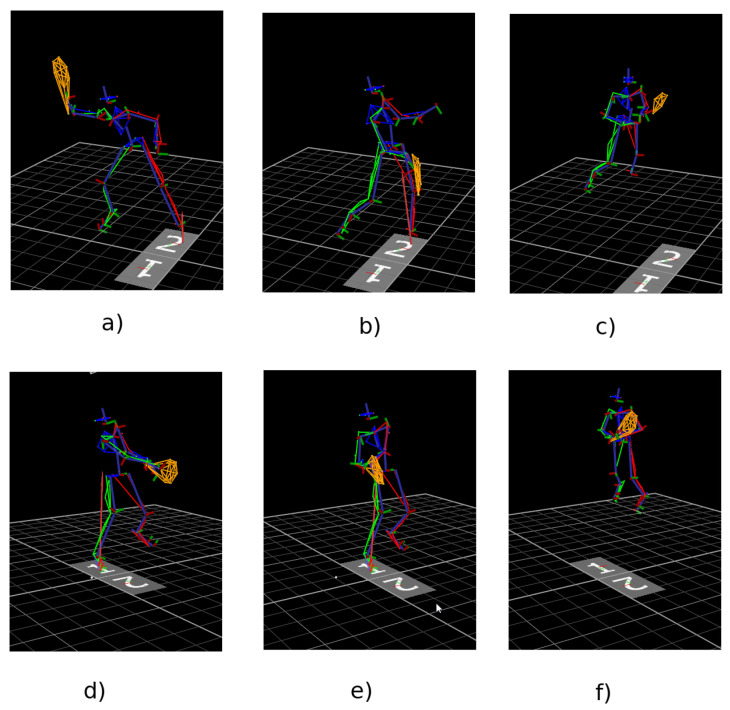
An example of forehand and backhand strokes (**a**) forehand preparation phase (**b**) forehand shot (**c**) no shot (**d**) backhand preparation phase (**e**) backhand shot (**f**) no shot.

**Figure 3 sensors-23-02422-f003:**
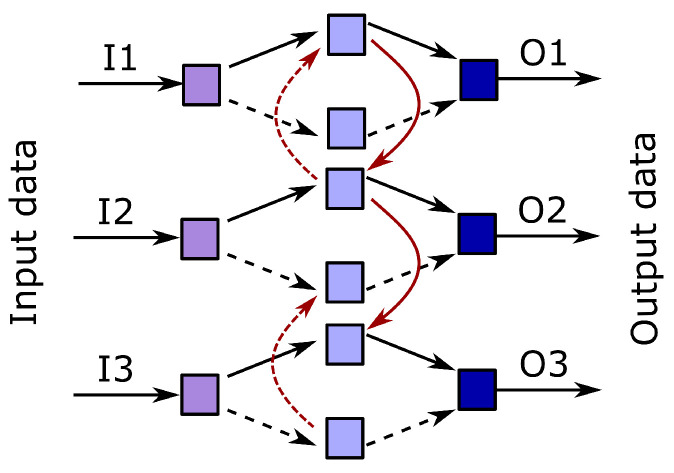
Scheme of the used BiRNN network.

**Figure 4 sensors-23-02422-f004:**
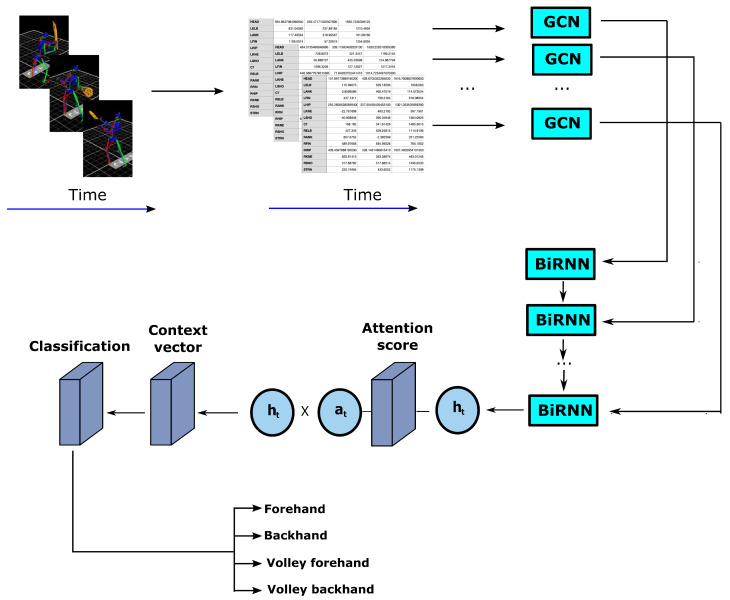
Classification model for tennis data movements.

**Figure 5 sensors-23-02422-f005:**
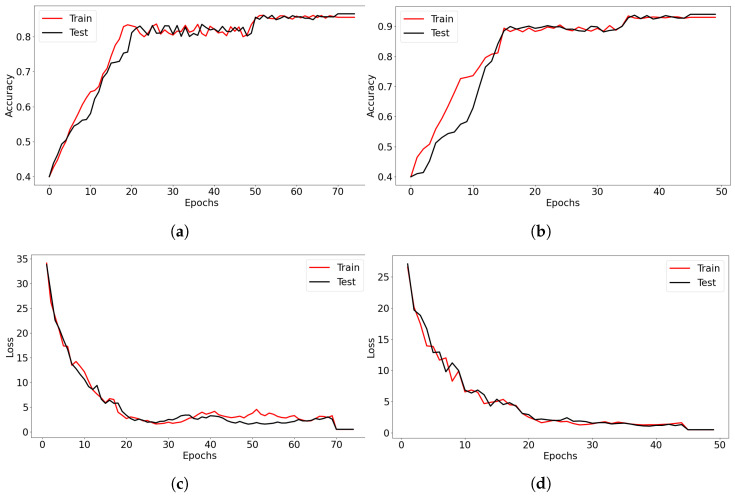
Selected learning parameters. (**a**) Learning accuracy for input without the tennis racket. (**b**) Learning accuracy for input with the tennis racket. (**c**) Loss value for input without the tennis racket. (**d**) Loss value for input with the tennis racket.

**Figure 6 sensors-23-02422-f006:**
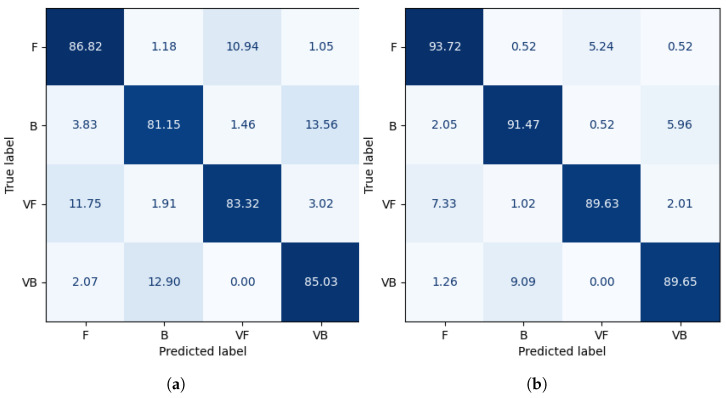
Confusion matrices (in %). (**a**) Matrix for input without the tennis racket. (**b**) Matrix for input with the tennis racket.

**Table 1 sensors-23-02422-t001:** Obtained Accuracy results for A3T-GCN.

Type of Input	Mean	Max	Min	±SD
Without racket	82.60%	85.54%	78.00%	2.08%
With racket	88.95%	93.00%	85.62%	2.62%

**Table 2 sensors-23-02422-t002:** Obtained Accuracies results for individual strokes.

Type of Input	Stroke	Mean	Max	Min	±SD
Without racket	Forehand	81.99%	85.54%	78.33%	2.36%
	Backhand	81.38%	85.25%	78.00%	2.49%
	Volley Forehand	82.72%	85.54%	78.26%	1.89%
	Volley Backhand	84.39%	85.43%	79.23%	2.02%
With racket	Forehand	89.84%	93.98%	85.89%	2.71%
	Backhand	88.41%	93.36%	85.71%	2.62%
	Volley Forehand	88.49%	93.87%	85.62%	2.82%
	Volley Backhand	89.08%	92.11%	86.60%	2.01%

**Table 3 sensors-23-02422-t003:** Obtained Precision results.

Type of Input	Stroke	Mean	Max	Min	±SD
Without racket	Forehand	86.99%	88.54%	82.98%	1.17%
	Backhand	80.99%	84.31%	76.41%	2.00%
	Volley Forehand	83.57%	85.85%	79.59%	2.00%
	Volley Backhand	84.50%	87.63%	78.00%	3.18%
With racket	Forehand	93.08%	97.89%	87.62%	4.38%
	Backhand	91.04%	94.90%	87.63%	2.34%
	Volley Forehand	89.45%	93.94%	85.00%	2.97%
	Volley Backhand	89.52%	93.00%	85.86%	2.28%

**Table 4 sensors-23-02422-t004:** Obtained Recall results.

Type of Input	Stroke	Mean	Max	Min	±SD
Without racket	Forehand	81.95%	85.00%	75.73%	2.95%
	Backhand	84.69%	87.76%	80.41%	2.37%
	Volley Forehand	87.22%	88.54%	83.87%	1.44%
	Volley Backhand	82.19%	85.00%	77.23%	2.22%
With racket	Forehand	89.09%	93.94%	85.00%	2.62%
	Backhand	89.71%	93.94%	85.86%	2.86%
	Volley Forehand	93.54%	97.89%	88.54%	3.96%
	Volley Backhand	90.76%	94.93%	86.73%	2.56%

**Table 5 sensors-23-02422-t005:** Obtained F1 results.

Type of Input	Stroke	Mean	Max	Min	±SD
Without racket	Forehand	84.39%	86.73%	79.19%	2.37%
	Backhand	82.77%	86.00%	78.39%	2.20%
	Volley Forehand	85.35%	87.18%	81.68%	1.17%
	Volley Backhand	83.33%	86.29%	77.61%	2.67%
With racket	Forehand	91.03%	95.88%	86.29%	3.38%
	Backhand	90.37%	94.42%	86.73%	2.58%
	Volley Forehand	91.44%	95.87%	86.73%	3.36%
	Volley Backhand	90.14%	93.48%	86.24%	2.40%

**Table 6 sensors-23-02422-t006:** Obtained Values for LOOCV.

Value	Input without Racket	Input with Racket
RMSE	9.64%	5.31%
SD RMSE	±5.58%	±5.79%

**Table 7 sensors-23-02422-t007:** Results comparison with the state-of-the-art. FH—forehand, BH—backhand, S—serve, BS—backspin, SM—smash, V—volley, H—hit, NH—no hit.

Data/Dataset	Type of Input	Classified Types of Tennis Movements	Detection Method	Accuracy	Paper
SensorTile	signal	FH	DNN	94–97%	[14]
		BH, S			
SensorTile	signal		SVM	90.82–98.86%	[15]
		FH,	NN	98.76–100%	
		BH,	DT	84.69–95.54%	
		FH, BH	RF	93.75–98.96%	
			kNN	87.76–99.44%	
IMU	sensor	FH, BH,	Pan Tompkins	80.6–98.1%	[16]
			algorithm		
		BS, S, SM			
THETIS	video	BH, V,	LSTM	81.23–89.42%	[19]
		FH, V, S, SM,			
THETIS	video	BH	SVM	51.20%	[17]
		FH, V, S, SM	CRF	86.44%	
THETIS	video	BH, V	Deep Historical	62%	[21]
		FH, V	LSTM		
HMDB51		S, SM		54%	
THETIS	video	BH	LSTM	70.17–97.67%	[22]
KTH		V, S, SM			
THETIS	video	BH	SVM	53.08–60.23%	[18]
		FH, V,			
KTH		S, SM		90.65%	
KTH	Video	S, H, NH	KLDA	73.34–92.29%	[23]
Broadcast	video	F, B	SVM	90.21%	[24]
Broadcast	video	F, B	SVM	87.10%	[25,26]
mixed	signal video	F, B, S	SVM,	89.69–97.02%	[27]
			kNN	95–98.67%	
			SVM	82.43–88.36%	
			kNN	89.41–93.44%	
			kNN	84.73–100%	
Vicon	images	FH, BH, NH	ST-GCN	86.3–87.3%	[28]
with fuzzy					
input					
Vicon	images	F, B, NH	ST-GCN	64.1–74.3%	[28]
Vicon	images	F, B, NH	A3T-GCN	86.9–93.82%	[30]
with fuzzy					
input					
Vicon	images	F, B, NH	A3T-GCN	74.22–81.95%	
Vicon	**3d**	F, B, V	A3T-GCN	**78.33–85.54**%	**Our**
	**silhouette**				**work**
Vicon	**3d**	F, B, V	A3T-GCN	**85.62–93.98**%	
	**silhouette**				
	**& racket**				

## Data Availability

Not applicable.

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
