# Peer review of "Temporal Pattern Attention for Multivariate Time Series of Tennis Strokes Classification"

_sensors, 2023, doi:10.3390/s23052422_

Round 1

Reviewer 1 Report

The paper describes how the inclusion of a tennis racket affects the classification of movement of stroke of a tennis player and the accuracy of using A3T-GCN based on motion capture data to classify the strokes.

# Language and editing

Line 3:

"understand the human" - unnecessary article

"behaviors" - as the journal is international, I would advise using British English. As this is a matter of decision rather than an error, I won't refer to each such occurence.

"identifying" - "identify"

"makes significant" - article missing

Line 4:

"for indicating" - "by indicating"

Line 5:

"how content" - article missing

"influences classification" - article missing

Line 6:

"An entirely player's silhoutte" - either entire silhouette or entirety of silhouette

Line 8:

"using motion capture system" - article missing

Line 14:

"whole-body" - "whole body"

Line 17:

"The Computer Vision" - unnecessary article

Line 18:

"the artificial intelligence" - unnecessary article

Line 22:

"in a time" - "at a time", or was it supposed to indicate a sequence?

Line 23:

"objects, (...) is involved" - "objects, (...) are involved"

Line 28:

"a detection" - incorrect article

Line 32:

"significant" - missing article

Line 33:

"recognition" - "and recognition"

Line 38:

"the most" - unnecessary article

Line 40:

"As an input" - unnecessary article

Line 41:

"As input parameters represented as basketball players measures like distances between joints and angles were applied" - unclear sentence

Line 42:

"Moreover, selection coordinates, depth maps together" - "Moreover, selection coordinates and depth maps together"

Line 44:

"analyse the karate" - "analysis of the karate"

Line 46:

"punch" - missing article

Line 49:

"for sports video" - "to sports video"

Line 52:

"dynamic movement" - missing article

Line 54:

"using Convolutional" - missing article

Line 55:

"The water skiing" - unnecessary article

Line 59:

"approach for analyzing" - "approach to analyzing"

"have been proposed" - "has been proposed"

Line 60:

"It took into the consideration" - unnecessary article

Line 62:

"embedded in spatial" - missing article

Line 64:

"has became" - "has become"

Line 65:

"every day" - everyday

Line 66:

"applied for recognition" - missing article

Line 75:

"the additional information" - unnecessary article

Line 78:

"ubasic" - "basic"

Line 84:

"This network also indicated the most important difference in parts to make a point of action recognition" - unclear, I assume it refers to the "Part-Attention", but how does that relate to a point of action recognition? And the difference between what?

Line 85:

"tennis" - "the tennis"

Line 90:

"backhand, slices" - "backhand, and slices"

Line 93:

"for classification" - "for the classification"

Line 94:

"strokes" - "stroke"

Line 95:

"very-well known" - "very well-known"

Line 102:

"the similar moves" - unnecessary article

Line 105:

"The transductive" - unnecessary article

"annotation" - missing article

Line 106:

"The changes of" - "The changes in"

Line 108:

"tennis serve" - "tennis serves"

Line 111:

"were used" - "was used"

Line 112:

"Forehand, backhands" - "Forehand and backhand"

Line 115:

"The Graph" - unnecessary article

Line 118:

"the recognition" - unnecessary article

Line 125:

"approach of tenis movements" - "approach to tenis movement"

Line 131:

"tennis strokes" - "tennis stroke"

Line 133:

"into how content" - missing article

Line 134:

"influence on classification" - "influences classification"

Line 138:

"introduces to the Attention" - "introduces the Attention"

It is this reviewer's opinion, that language and editing errors present in Abstract and 1. Introduction, as listed above, are representative of the entire paper. While those are quite numerous, they are mostly easy to fix, even using automatic tools. Nevertheless, it is this reviewer's opinion that those should be corrected before final publication.

# Presentation problems

Equation 1 uses undefined symbols: n, Theta (it is mentioned, but not defined). Also, the size of fractions is almost the size of symbols, which might be confusing. I would advise using decimal fractions in this particular case. The same goes for line 189.

Further in the text, the O is not defined, F is only said to be related.

In equations 3-6, the W_u, W_r, W_c are not defined,

In equation 7, H is not defined.

While those symbols might be understood by a reader depending on previous knowledge, the paper should not rely on it to solve the ambiguity.

# Other problems

Chapter 2 describes the setup of the experiments. It is clear, that Vicon Nexus was used, however, it remains unclear how many cameras were involved, what were the angles between the cameras (horizontal and vertical), and what were the calibration errors. The chapter mentions that interpolation was used to fill the gaps, which is natural, but without the information about the setup of the cameras, one cannot estimate the quality of the data and therefore cannot say whether interpolation significantly affected the results or was it just a matter of having proper continuous recordings. This is especially important in terms of the next comment.

Chapter 4 clearly indicates that the results with the racket are better than without. Since the racket is a rigid body, however, the 7 markers used to represent it should move in unison. While the number of markers might be useful, due to technical constraints (occlusion), it remains unclear, whether the improvement is a result of a better description of wrist movement, or is it a matter of movement of racket inside of a hand, due to changes in grip. As authors use Plugin GAIT, calculations of rotation in the wrist are performed on the basis of virtual markers, the position of which is based on real markers. In the standard 39-markers template, only a single marker is dependent on the rotation in the wrist. It can be easily occluded, especially in the case, where an actor has a racket. It is therefore fully possible, that markers on the racket are actually a representation of rotation in the wrist. It could be tested with additional markers on hand, but that would require additional recordings. One could, however, remove the information about the wrist, and see how this affects the result, also, the racket could be represented as a virtual marker as well.

Chapter 4 presents the values of binary classification in a multinomial classification problem. It is unclear whether in cases where Backhand was classified incorrectly, it was the second best option in ranked order, or was it the last. I would advise using aggregation over ranks in order to present this information, however, I do not believe that this is strictly necessary for the paper to be published.

The training process remains undescribed. As such, it cannot be determined whether the architecture used achieved the peak of its possibilities or whether can it be further improved by changing the training procedure.

As the dataset nor the code is published, the results cannot be reproduced.

Author Response

Dear Reviewer,

thank you for all detailed comments and suggestions. We found them very useful as we wrote our revision.

Please find below, in a file,  our responses.

Kind regards,

Authors

Reviewer 2 Report

There are multiple shortcomings in the paper:

1) Figures depiciting recorded actions would be useful to better understand the analyzed motion

2) There is mention that one of the subjects was left-handed but there is no indication how this was handeled. Was the data mirrored? Or was the GCN supposed to handle this? 

3) Was the quality of performed action considered? Is there much variance in the way of performing the action between subjects and between repetitions of the same subject? Could it have impact on the results?  Without more extensive description it is difficult to understand the nature of the problem.

4) It is very unclear what is the authors contributon to the proposed method, which is based on the A3T-GCN. In Section 3 it says: 

- 'basic modification of this consists of transforming the element responsible for the prediction into a classifier', which a technicality, not a novelty; 

- 'Additionally, the elements responsible for the separation of spatial and temporal features have also been adapted' - it is unclear what elements exactly have beed modified and how - what's the actual difference with the original method?

5) The paper lacks any comparision to any other method (!), including the original A3T-GCN method, which is needed to evaluate the proposed modifications (which are unclear - see 4). There's only a simple mention that 'The mean accuracy obtained in this study is higher for both the analyzed silhouette and its combination with a tennis racket compared to the results 
obtained in the work [28]' - without any numerical values (!), not to mention that the referenced method is authors' previous work and therefore there is no mention of any external SOTA method (of which there are plenty for HAR)

6) Experiments should be performed using leave-one-out cross-validation, otherwise it is impossible to say if the model is not overfitting and what's it's use in practical scenario.

7) Why the results in Table 1 (accuracy) are not given per-action, while all other metrics are?

8) The paper lacks proper discussion of results. Why are the results so poor? There are only 4 actions and intuitively those are quite easy to distinguish having precise mocap data - it seems to be sufficient to distinguish forward/backward motion (forehand vs backhand) and measure distance between hands (one-handed vs two-handed grip). My understading of this motion may be very simplified and incorrect - but there is no explanation in the paper that would help me understand it better (see also 3). Please also consider this in the context of other (public) datasets - e.g. for the NTU-RGBD+D dataset (referenced in the paper), which includes 60 actions (!) better results are obtained in literature using Kinect data instead of high-precision mocap.

9) In conclusions we can read that 'The collected results clearly indicate that the use of the A3T-GCN structure is the most suitable for motion data expressed in a form of c3d files' - results indicate no such thing, as there is no comparision with other methods.

Author Response

(The authors gave the same response as above.)

Reviewer 3 Report

The authors used the Attentional Temporal Graph Convolutional Network (A3T-GCN) to classify tennis strokes and found that classification accuracies were numerically higher when player and racket silhouettes served as input data compared to when only player silhouettes were used.

The structure of the manuscript does not conform to the standard used in scientific journals. The authors need to restructure the manuscript to include the following sections: introduction, materials and methods (including subsections on participants, data collection, data preprocessing and analyses etc.), results, discussion and conclusion. Even more problematic, though, is the fact that the authors didn’t apply inferential statistical analyses to their classification results. These and other issues will have to be addressed before the manuscript can be considered for publication.

1 Introduction

The introduction lists a lot of studies, but it remains unclear how they relate to the present study and its rationale. Some substantial restructuring is required. In particular, the introduction should address the following questions (among others): Why is A3T-GCN suitable for classifying tennis stroke data, and what are its potential advantages compared to other classification methods that have been used in previous research? Why were forehand, backhand and volley strokes chosen to be classified? Which analysis basis (player coordinates vs. player+racket coordinates) has previously yielded more reliable classification results?

2 Acquisition of tennis strokes

This section, as well as section 3 and parts of section 4 should be subsumed under a new section labeled “materials and methods”.

The materials and methods section needs to be much more detailed. The authors need to add more participant information (age, height, weight, years of tennis experience/level of professionalisation etc.). Biometric data in particular is necessary to estimate how far-reaching the results of the present study are, i.e. whether A3T-GCN can successfully classify tennis strokes for a variety of body types.

The authors need to clarify how many tennis strokes of each type each participant performed. It is also unclear whether all subjects performed both a one-handed and a two-handed backhand stroke or whether they performed only one type (according to their personal preferences). If the latter is true, how many participants performed the two-handed as opposed to the one-handed version and did the type of backhand performed affect categorisation accuracy?

Recording post-processing needs to be described in much more detail, particularly the interpolation methods used for gap filling need to be explained.

The authors state that “the whole gathered recordings was verified by a professional tennis coach” (p.4). In what way were the data verified? Did they rate the strokes for accuracy of execution?

Another minor question I have is why participants were asked to warm-up before marker attachment? Marker attachment would presumably have been time-consuming and participants would have to warm up again before performing the actual tennis strokes?

4 Experiments and results

Descriptions of the experiments and classifiers should be removed from the results section and put in the materials and methods section.

The authors need to apply inferential statistical analyses to their classification results; just comparing numerical values in tables 1-4 is not a sufficient basis for the claim that classification results for data with a tennis racket are superior to those from data without the racket.

The authors state that of the 10 participants they examined one was female and one was left-handed. Did gender or handedness of the observed player affect the classification results in any way?

Additional points

Figure 1: The subtitle needs to include a description of what is depicted in the figure and a key to the abbreviations used in the figure.

The manuscript should be proofread by a native speaker.

Author Response

(The authors gave the same response as above.)

Round 2

Reviewer 2 Report

The authors have addressed some of the issues but not all of them. Some major improvements are still needed:

1) The only included numerical comparison is with a previous paper, in which different classes were considered (!). Such comparison is incorrect. Moreover, it is not even clear if this was the same dataset. 

2) The proposed classification method is still not compared to any other method, therefore there is no justification for using this particular method, nor for using modifications introduced by the authors. 

3) Regarding my previous comment: 

'Experiments should be performed using leave-one-out cross-validation, otherwise it is impossible to say if the model is not overfitting and what's it's use in practical scenario.'

authors responed:

'Thank you for drawing attention to this aspect. The information about the learning process has been added (see Fig. 4)'

which does not address the raised issue at all. There is no answer regarding LOO cross-validation nor any analysis of overfitting due to using same subjects in training and test sets (which probably happend due to random division of training/val/test sets).

 4) Regarding contribution of the paper the authors wrote:

'In the original approach the Gated Recurrent Unit (GRU) consisted of LSTM network was applied. In this study the Recurrent Neural Network was used'

This is incorrect, as both GRU and LSTM are types of RNN. Not to mention that in Fig. GRU is presented.

5) I appreciate that the authors explained in their response why it is difficult to distinguish some of the considered actions, however this explanation should be included in the paper as well.

Author Response

Dear Reviewer,

thank you for all detailed comments and suggestions. We found them very useful as we wrote our revision.

Please find our responses in the file.

 Kind regards,

Authors
